# Scaling Up Differentially Private LASSO Regularized Logistic Regression via Faster Frank-Wolfe Iterations

**Edward Raff**[*]
Booz Allen Hamilton
raff_edward@bah.com

**Amol Khanna**[*]
Booz Allen Hamilton
khanna_amol@bah.com

**Fred Lu**
Booz Allen Hamilton
lu_fred@bah.com

## Abstract

To the best of our knowledge, there are no methods today for training differentially private regression models on sparse input data. To remedy this, we adapt the Frank-Wolfe algorithm for $L_1$ penalized linear regression to be aware of sparse inputs and to use them effectively. In doing so, we reduce the training time of the algorithm from $\mathcal{O}(TDS + TNS)$ to $\mathcal{O}(NS + T\sqrt{D}\log D + TS^2)$, where $T$ is the number of iterations and a sparsity rate $S$ of a dataset with $N$ rows and $D$ features. Our results demonstrate that this procedure can reduce runtime by a factor of up to $2,200\times$, depending on the value of the privacy parameter $\epsilon$ and the sparsity of the dataset.

## 1 Introduction

Differential Privacy (DP) is currently the most effective tool for machine learning practitioners and researchers to ensure the privacy of the individual data used in model construction. Given parameters $\epsilon$ and $\delta$, on any two datasets $\mathcal{D}$ and $\mathcal{D}'$ differing on one example, an (approximately) differentially private randomized algorithm $\mathcal{A}$ satisfies $\Pr\left[\mathcal{A}(\mathcal{D}) \in O\right] \leq \exp\{\epsilon\}\Pr\left[\mathcal{A}(\mathcal{D}') \in O\right] + \delta$ for any $O \subseteq \text{image}(\mathcal{A})$ [1]. Note that lower values of $\epsilon$ and $\delta$ correspond to stronger privacy.

While DP has had many successes in industry and government, DP-based machine learning methods have made little progress for sparse high-dimensional problems [2, 3, 4, 5]. We believe that this issue arises because, to the best of our knowledge, given a dataset with $D$ features and a training algorithm with $T$ iterations, all current iterative DP regression algorithms require at least $\mathcal{O}(TD)$ training complexity, as shown in Table 1. This makes it impractical to use these algorithms on any dataset with a large number of features. *Our solution to this problem is to take already existing algorithms, and remove all redundant computations with mathematically equivalent steps*. This ensures that, by construction, we retain all proofs of correctness — but end with a faster version of the same method.

Table 1: A summary of prior methods for solving $L_1$ regularized Logistic Regression, which do not take advantage of sparsity in the input data.

| Method | Complexity |
|---|---|
| Frank-Wolfe Methods [6, 7, 8, 9] | $\mathcal{O}(TND)$ |
| ADMM [10] | $\mathcal{O}(TNDM)$ |
| Iterative Gradient Hard Thresholding Methods [9, 11, 12] | $\mathcal{O}(TND)$ |
| Coordinate Descent [13] | $\mathcal{O}(TND)$ |
| Mirror Descent [8] | $\mathcal{O}(TNDM)$ |

---

[*]Co-first authors for equal contribution.

37th Conference on Neural Information Processing Systems (NeurIPS 2023).

We are interested in creating a differentialy private machine learning algorithm which scales to sparse datasets with high values of $D$, so we look toward the LASSO regularized logistic regression model [14]. Specifically, given a dataset $\{\mathbf{x}_1, \ldots, \mathbf{x}_N\} \in \mathbb{R}^D$ which can be represented as a design matrix $X \in \mathbb{R}^{N \times D}$, $\{y_1, \ldots, y_N\} \in \{0, 1\}$, and a maximum $L_1$ norm $\lambda$, we wish to solve

$$\widehat{\mathbf{w}} = \underset{\mathbf{w} \in \mathbb{R}^D: \|\mathbf{w}\|_1 \leq \lambda}{\arg \min} \frac{1}{N} \sum_{i=1}^{N} \mathcal{L}(\mathbf{w} \cdot \mathbf{x}_i) \tag{1}$$

where $\mathcal{L}(\cdot)$ is the loss function. In this paper, we will use the logistic loss to avoid exploiting any closed-form updates/solutions in the linear case, but our results are still applicable to linear regression. To do so with DP, we use the Frank-Wolfe algorithm as it is well studied for $L_1$ constrained optimization and regularly used in DP literature [15, 16]. Frank-Wolfe is also desirable because when properly initialized, its solutions will have at most $T$ nonzero coefficients for $T$ training iterations. Though DP noise can reduce accuracy and cause increased density of solutions through suboptimal updates, when considering problems with more than 1 million features, we are unlikely to perform 1 million iterations, so a benefit is still obtained [17].

We develop the first sparse-dataset friendly Frank-Wolfe algorithm to remediate the problem of no sparse algorithms for high-dimensional DP regression. Because a sparse-efficient Frank-Wolfe algorithm does not exist today even for non-private problems, our work proceeds in three contributions:

1. We analyze the numerical updates of the Frank-Wolfe algorithm to separate the task into (a) "queue maintenance" for determining the next coordinate to update and (b) sparse updates of the solution and intermediate variables. If the average column sparsity of $X$ is $S_c < D$ and the average row sparsity of $X$ is $S_r < N$, we show for the first time that Frank-Wolfe can update its solution in $S_r S_c$ work per iteration.

2. We show that in the non-private case, the queue to select the next coordinate can be maintained in $\mathcal{O}(\|\mathbf{w}\|_0 \log D)$ time, but is cache-unfriendly.

3. Finally, we develop a new Big-Step Little-Step Sampler for DP Frank-Wolfe that can be maintained and sampled from in $\mathcal{O}(\sqrt{D} \log D)$ time.

We test our algorithm on high-dimensional problems with up to 8.4 million datapoints and 20 million features on a single machine and find speedups ranging from $10\times$ to $2,200\times$ that of the standard DP Frank-Wolfe method. Critically, our approach is mathematically equivalent, and so retains all prior proofs of correctness.

The remainder of our work is organized as follows. In section 2 we will review related work in more detail and discuss the lack of sparse dataset algorithms for Frank-Wolfe and DP. Next, we develop the nonprivate and DP variants of our sparse-friendly Frank-Wolfe in section 3. In section 4 we demonstrate the empirical correctness of our algorithm, a speedup in runtime for the DP case, and new state-of-the-art accuracy in high-dimensional logistic regression. Finally, we conclude in section 5.

## 2 Related Work

As best as we can find, no works have specifically considered a sparse-dataset efficient Frank-Wolfe algorithm. Additionally, we find that work studying $L_1$-regularized DP regression has not reached any high-dimensional problems. Our work is the first to show that Frank-Wolfe iterations can be done with complexity sub-linear in $D$ for sparse datasets. It also produces a sparse weight vector.

For DP regression, the addition of noise to each variable has limited exploration of sparse datasets. Most works study only one regression problem with less than 100 dense variables or are entirely theoretical with no empirical results [11, 10, 6, 18, 19]. To the best of our knowledge, the largest scale attempt for high-dimensional logistic regression with DP is by Iyengar et al., who introduce an improved objective perturbation method for maintaining DP with convex optimization [16]. This required using L-BFGS, which is $\mathcal{O}(D)$ complexity for sparse data and produces completely dense solution vectors $\mathbf{w}$ [20]. In addition, Wang & Gu attempted to train an algorithm on the RCV1 dataset but with worse results and similar big-$\mathcal{O}$ complexity to Iyengar et al. [11]. While Jain & Thakurta claim to tackle the URL dataset with 20M variables, their solution does so by sub-sampling just 0.29% of the data for training and 0.13% of the data for validation, making the results suspect and

non-scalable [21]. Lastly, a larger survey by Jayaraman & Evans shows no prior work considering the sparsity of DP solutions and all other works tackling datasets with less than 5000 features [22]. In contrast, our work does no sub-sampling and directly takes advantage of dataset sparsity in training high-dimensional problems. Our use of the Frank-Wolfe algorithm also means our solution is sparse, with no more than $T$ non-zero coefficients for $T$ iterations.

In addition to no DP regression algorithm handling sparse datasets, we cannot find literature that improves upon the $\mathcal{O}(D)$ dependency of Frank-Wolfe on sparse data. Many prior works have noted this dependence as a limit to its scalability, and column sub-sampling approaches are one method that has been used to mitigate that cost [23, 15, 24]. Others have looked at distributed Map-Reduce implementations [25] or adding momentum terms [26] as a means of scaling the Frank-Wolfe algorithm. However, our method is the first to address dataset sparsity directly within the Frank-Wolfe algorithm, and can generally be applied to these prior works with additional derivations for new steps. Other methods that apply to standard regression, use the $L_2$ penalty [27, 28] would require modification to use our approach. Similarly, pruning methods [29] may require adaption to account for the privacy impact.

Of particular note is [30] which tackles sub-linear scaling in the number of rows $N$ when $N > D$, but is primarily theoretical. Their work is the first to introduce the idea of "queue maintenance" that we similarly leverage in this work. Despite a conceptually similar goal of using a priority queue to accelerate the algorithm, [30] relies on maximum inner product search, where our queues are of a fundamentally different structure.

To the best of our knowledge, the COPT library is the only Frank-Wolfe implementation that makes any effort to support sparse datasets but is still $\mathcal{O}(D)$ iteration complexity, so we base our comparison against its approach [31]. No DP regression library we are aware of supports sparse datasets [32].

## 3 Methods

Throughout this section, sparsity refers to an algorithm's awareness and efficiency of handling input data which contains mostly zero values. We will first review the only current method for using Frank-Wolfe with sparse data and establish its inefficiency. We will then detail how to produce a generic framework for a sparse dataset efficient Frank-Wolfe algorithm using a queuing structure to select the next coordinate update. Having established a sparse friendly framework, we will show how to obtain $\mathcal{O}(NS_c + T\|\mathbf{w}^*\|_0 \log D + TS_rS_c)$ complexity for the non-private Frank-Wolfe algorithm by using a Fibonacci heap, where $\mathbf{w}^*$ is the solution at convergence. Then we replace the Fibonacci heap with a sampling structure to create a DP Frank-Wolfe algorithm with $\mathcal{O}(NS_c + T\sqrt{D} \log D + TS_rS_c)$ complexity.

### 3.1 Frank-Wolfe Iterations in sub-$\mathcal{O}(D)$ Time

The only work we could find of any sparse-input aware Frank-Wolfe implementation is the COPT library, which contains two simple optimizations: it pre-computes a re-used dense vector (i.e., all values are non-zero) and it uses a sparse matrix format for computing the vector-matrix product $X\mathbf{w}$ [31]. The details are abstracted into Algorithm 1, where each line has a comment on the algorithmic complexity of each step. Note to make the algorithm DP, we have added a $+\text{Lap}\left(\frac{\lambda L\sqrt{8T\log(1/\delta)}}{N\epsilon}\right)$ to draw noise from a zero-mean Laplacian distribution with the specified scale, where $\lambda$ is the constraint parameter and $L$ is the $L_1-$Lipschitz constant of the loss function $\mathcal{L}(\cdot)$. If a non-private Frank-Wolfe implementation is desired, this value can be ignored.

In this and other pseudo-codes, we explicitly write out all intermediate computations as they are impor-

**Algorithm 1** Standard Sparse-Aware Frank-Wolfe

1: $\mathbf{w}_0 \leftarrow \mathbf{0}$
2: $\bar{\mathbf{y}} \leftarrow X^\top \mathbf{y}$ $\qquad\qquad\qquad \mathcal{O}(NS_c)$
3: **for** $t = 1$ to $T - 1$ **do**
4: $\quad \bar{\mathbf{v}}_t \leftarrow X\mathbf{w}_t$ $\qquad\qquad \mathcal{O}(NS_c)$
5: $\quad \bar{\mathbf{q}}_t \leftarrow \nabla\mathcal{L}(\bar{\mathbf{v}}_t)$ $\qquad\qquad \mathcal{O}(N)$
6: $\quad \bar{\mathbf{z}}_t \leftarrow X^\top \bar{\mathbf{q}}_t$ $\qquad\qquad \mathcal{O}(NS_c)$
7: $\quad \boldsymbol{\alpha}_t \leftarrow \bar{\mathbf{z}}_t - \bar{\mathbf{y}}$ $\qquad\qquad \mathcal{O}(D)$
8: $\quad j \leftarrow$
$\quad \arg\min_j \left|\alpha_t^{(j)} + \text{Lap}\left(\frac{\lambda L\sqrt{8T\log\frac{1}{\delta}}}{N\epsilon}\right)\right|$
$\quad \mathcal{O}(D)$
9: $\quad \mathbf{d}_t = -\mathbf{w}_t$ $\qquad\qquad\qquad \mathcal{O}(D)$
10: $\quad d_t^{(j)} \leftarrow d_t^{(j)} - \lambda \cdot \text{sign}\left(\alpha_t^{(j)}\right)$ $\quad \mathcal{O}(1)$
11: $\quad g_t = -\langle \boldsymbol{\alpha}_t, \mathbf{d}_t \rangle$ $\qquad\qquad \mathcal{O}(D)$
12: $\quad \eta_t = \frac{2}{t+2}$ $\qquad\qquad\qquad \mathcal{O}(1)$
13: $\quad \mathbf{w}_{t+1} = \mathbf{w}_t + \eta_t\mathbf{d}_t$ $\qquad\qquad \mathcal{O}(D)$
14: **end for**
15: Output $\mathbf{w}_T$

tant for enabling sparse updates. $\bar{\mathbf{y}}$ is an intermediate variable for the labels in the gradient of a linear problem that is pre-computed once and reused. $\bar{\mathbf{z}}$ is a temporary variable. $\bar{\mathbf{q}}$ and $\boldsymbol{\alpha}$ are the gradients with respect to each row and column respectively. $\bar{\mathbf{v}}$ is the dot product of each row with the weight vector, and $\mathbf{d}$ is the update direction. The iteration subscript $t$ will be dropped when unnecessary for clarity. The superscript $^{(j)}$ denotes updating the $j$'th coordinate of a vector and leaving others unaltered.

While lines 2, 4, and 6 of the algorithm exploit the sparsity of the data, this only reduces the complexity to $\mathcal{O}(NS_c)$ plus an additional dense $\mathcal{O}(D)$ work for lines 7 through 13 and $\mathcal{O}(N)$ work for line 5. This results in a final complexity of $\mathcal{O}(TNS_C + TD)$. For high dimensional problems, especially when $N \ll D$, this is problematic in scaling up a Frank-Wolfe based solver.

To derive a Frank-Wolfe algorithm that is more efficient on sparse datasets, we will assume there is an abstract priority queuing structure $Q$ that returns the next coordinate to update $j$ for each iteration. We will detail how to design a $Q$ to use this algorithm in non-private and DP cases in the following two sections.

### Sparse $\mathbf{w}_t$ Updates

In line 13 of Algorithm 1, if we ignore the change to coordinate $j$ of $\mathbf{d}_t$, we can write $\mathbf{w}_{t+1} = \mathbf{w}_t - \eta_t \mathbf{w}_t$, which can be re-written as $\mathbf{w}_{t+1} = (1 - \eta_t)\mathbf{w}_t$. If we represent the weight $\mathbf{w}_{t+1} = \mathbf{w} \cdot w_m$ with a co-associated multiplicative scalar $w_m$, we can alter $w_m \leftarrow w_m \cdot (1 - \eta_t)$ to have the same effect as altering all $D$ variables implicitly. Then the $j^{\text{th}}$ coordinate can be updated individually, allowing line 13 of Algorithm 1 to run in $\mathcal{O}(1)$ time. We will use the same trick to represent $\bar{\mathbf{v}}_t = \bar{\mathbf{v}} \cdot w_m$ as it has the same multiplicative scale.

### Sparse $\boldsymbol{\alpha}$ and $\bar{\mathbf{v}}$ Updates

The dot product scores $\bar{\mathbf{v}}_t$ and column gradients $\boldsymbol{\alpha}_t$ are intrinsically connected in their sparsity patterns. When the $j^{\text{th}}$ value of $\mathbf{w}_t$ is altered, multiple values of $\bar{\mathbf{v}}_t$ change, which propagates changes to the gradients $\boldsymbol{\alpha}_t$. However, the elements $\{i\}$ of $\bar{\mathbf{v}}_t$ that change are only those where rows $\{i\}$ in $X$ use feature $j$. Let $\tilde{d}_j$ represent a perturbation to the $j$'th update direction. Each row $i$ that uses the $j$'th feature will then be alter the variable $\bar{v}^{(i)}$ by $-\eta_t \tilde{d} X[i, j]$. This lets us handle line 10 sparsely.

Each row $\{i\}$ that changes in $\bar{\mathbf{v}}_t$ propagates to the values in $\bar{\mathbf{q}}$. We can represent the change in gradient value between iterations as $\gamma$, which can then be used to sparsely update the $\boldsymbol{\alpha}$ values by noting that $X^\top \bar{\mathbf{q}}$ would change only by the non-zero columns of $X[i, :]$. So we can compute the update $\boldsymbol{\alpha}$ by $\gamma \cdot X[i, :]$. By updating $\boldsymbol{\alpha}$ directly, we do not need to account for the contribution of $\bar{\mathbf{y}}$ after the first iteration.

### Sparse $g_t$ Updates

The final variable of interest is the Frank-Wolfe convergence gap $g_t = -\boldsymbol{\alpha}_t^\top \mathbf{d}_t$. Instead of recomputing this every iteration, we can keep a base value $\tilde{g}$ that is altered based on both of the prior two insights. When $w_m$ is updated, we re-scale $\tilde{g}$ by $(1 - \eta_t)$ and add $\eta_t \tilde{d} \alpha^{(j)}$, the change in the dot product

---

**Algorithm 2** Fast Sparse-Aware Frank-Wolfe for Linear Models

1: $\mathbf{w} \leftarrow \mathbf{0}$
2: $w_m \leftarrow 1$
3: $\tilde{g} \leftarrow 0$
4: $\bar{\mathbf{y}} \leftarrow X^\top \mathbf{y}$   $\mathcal{O}(NS_c)$
5: $scale \leftarrow \frac{LN\epsilon}{2\lambda\sqrt{8T \log(1/\delta)}}$
6: $Q \leftarrow$ Priority Queue or Sampling Algorithm
7: **for** $t = 1$ to $T - 1$ **do**
8:   **if** $t = 1$ **then**
9:    $\bar{\mathbf{v}} \leftarrow Xw$   $\mathcal{O}(NS_c)$
10:    $\bar{\mathbf{q}} \leftarrow \nabla\mathcal{L}(\bar{\mathbf{v}})$   $\mathcal{O}(N)$
11:    $\bar{\mathbf{z}} \leftarrow X^\top \bar{\mathbf{q}}$   $\mathcal{O}(NS_c)$
12:    $\boldsymbol{\alpha} \leftarrow \bar{\mathbf{z}} - \bar{\mathbf{y}}$   $\mathcal{O}(D)$
13:    $Q.\text{add}(j, |\alpha^{(j)}| \cdot scale) \ \forall \ j \in \{1, \dots, D\}$   $\mathcal{O}(D)$
14:   **end if**
15:   $j \leftarrow Q.\text{getNext}()$   Select coordinate to update
16:   $\tilde{d} \leftarrow -\lambda \cdot \text{sign}\left(\alpha^{(j)}\right)$   $\mathcal{O}(1)$
17:   $g_t \leftarrow \tilde{g} - \tilde{d} \cdot \alpha^{(j)}$   $\mathcal{O}(1)$
18:   $\eta_t = \frac{2}{t+2}$   $\mathcal{O}(1)$
19:   $w_m \leftarrow w_m(1 - \eta_t)$
20:   $w^{(j)} \leftarrow w^{(j)} \eta_t \tilde{d} / w_m$
21:   $\tilde{g} \leftarrow \tilde{g}(1 - \eta_t) + \eta_t \tilde{d} \alpha^{(j)}$
22:   **for all** rows $i$ of $X$ with feature $j$ **do** $\mathcal{O}(S_r)$
23:    $\bar{v}^{(i)} \leftarrow \bar{v}^{(i)} + \eta_t \tilde{d} * X[i, j] / w_m$   $\mathcal{O}(1)$
24:    $\gamma \leftarrow \nabla\mathcal{L}(w_m \cdot \bar{v}^{(i)}) - \bar{q}^{(j)}$   $\mathcal{O}(1)$
25:    $\bar{q}^{(j)} \leftarrow \bar{q}^{(j)} + \gamma$   $\mathcal{O}(1)$
26:    $\boldsymbol{\alpha} \leftarrow \boldsymbol{\alpha} + \gamma \cdot X[i, :]$   $\mathcal{O}(S_c)$
27:    $\tilde{g} \leftarrow \tilde{g} + \gamma \cdot X[i, :]^\top \mathbf{w} \cdot w_m$   $\mathcal{O}(S_c)$
28:   **end for**
29:   $Q.\text{update}(k, |\alpha^{(k)}| \cdot scale) \ \forall \ k$ gradients updated
30: **end for**
31: Output $\mathbf{w}$

---

$-\boldsymbol{\alpha}_t^\top \mathbf{d}_t$ caused by just the $j^{\text{th}}$ coordinate update. After the sparse $\boldsymbol{\alpha}$ updates, $\tilde{g}$ is again updated by $\gamma X[i,:]^\top \mathbf{w} \cdot w_m$.

**Fast Frank-Wolfe Framework**

The final procedure that forms the foundation for our results is given in Algorithm 2. The *scale* variable holds the noise parameter required for DP and can be ignored for non-private training. Lines 6, 13, 15, and 30 require an abstract priority queue that is populated with values proportional each feature's gradient. This mechanism is different in the non-private and private cases, and we will detail them in the following two sections.

The first iteration of Algorithm 2 performs the same calculations as Algorithm 1, but for all subsequent iterations, values will be updated in a sparse fashion.

Lines 16-21 update the multiplicative $w_m$ variables and perform the single coordinate updates to $\mathbf{w}$ and $\tilde{g}$, all taking $\mathcal{O}(1)$ time to complete. Lines 22-29 handle the updates for $\boldsymbol{\alpha}$ and $\bar{\mathbf{v}}$, which requires looping over the rows that use feature $j$, which we expect to have $\mathcal{O}(S_r)$ complexity. Within the loop, we use one row of the matrix to perform a sparse update which we expect to have $\mathcal{O}(S_c)$ complexity. The gradients $\alpha^{(k)}$ that get updated by this loop are updated in the priority queue $Q$ [2] in $\mathcal{O}(1)$ time per update, so they do not alter the final complexity. The final complexities of our algorithm are now dependent on the complexity of $Q.\texttt{getNext}()$, which will differ in the non-private and private cases.

## 3.2 Algorithmically Efficient Non-Private Frank-Wolfe

We first analyze the complexity of the non-private Frank-Wolfe algorithm, though our ultimate goal is to make the private case more efficient. The algorithm we detail in the non-private case will be of superior big-$\mathcal{O}$ complexity and perform significantly less FLOPs than the standard Frank-Wolfe implementation but will not be faster in practice due to constant factor overheads which we will explain.

The primary insight in building an algorithmically-faster non-private algorithm is to use a Fibonacci Heap, which allows for $\mathcal{O}(\log D)$ removal complexity and amortized $\mathcal{O}(1)$ insertion and `decreaseKey` operations. We use the negative magnitude as the key in the min-heap. Our insight is that we can decrease a key $j$ whenever $|\alpha^{(j)}|$ increases, and ignore cases where $|\alpha^{(j)}|$ decreases. This means the negative priority is an upper bound on true gradient magnitude. This is favorable because the vast majority of updates to the queue are intrinsically of a magnitude too small to be selected (hence why the solution is sparse) and so even with an inflated magnitude are never top of the queue.

These stale gradients will cause some items to reach the top of the queue incorrectly, which is easy to resolve as detailed in Algorithm 3. The current item $c$ is popped off the queue, and compared against the current best coordinate $j$. This loop continues until the top of the queue has a smaller priority than the current gradient magnitude $\alpha^{(j)}$. Because the stale

---

**Algorithm 3** Fibonacci Heap Frank-Wolfe Queue Maintenance

1: **function** GETNEXT( )
2:    $j \leftarrow -1$    Assume $|\alpha^{(-1)}|$ returns $-\infty$
3:    **repeat**
4:      $c \leftarrow Q.\texttt{pop}()$         $\mathcal{O}(\log D)$
5:      **if** $|\alpha^{(c)}| > |\alpha^{(j)}|$ **then**
6:        $j \leftarrow c$
7:      **end if**
8:    **until** $|\alpha^{(j)}| > |Q.\texttt{peekPriority}()|$
9:    Re-insert removed items $c_{...}$ using priorities $|\alpha^{(c...)}|$
10:    Output $j$
11: **end function**

12: **function** UPDATE$(i, v)$
13:    $v_{cur} \leftarrow$ Current priority of item $i$
14:    **if** $-v_{cur} > -v$ **then**      Min-Heap
15:      $Q.\texttt{decreaseKey(i, }v)$      $\mathcal{O}(1)$
16:    **end if**
17: **end function**

---

magnitude can only be larger than the true gradient, once we satisfy this condition it must be the case that no item in the queue can have a higher priority. Thus, the procedure is correct.

The number of items we expect to have to consider must be proportional to the number of non-zero coefficients in the weight vector. This gives a final complexity of `getNext` $\mathcal{O}(\|\mathbf{w}^*\|_0)$ for the number of non-zeros in the final solution, multiplied by the $\mathcal{O}(\log(D))$ cost per `pop()` call, giving a final complexity of $\mathcal{O}(NS_c + T\|\mathbf{w}^*\|_0 \log D + TS_r S_c)$.

---

[2]There are multiple ways to implement this, but we find a naïve re-iteration over the loop to update based on the final values the fastest due to reduced memory overheads.

While this is of superior algorithmic complexity compared to the standard Frank-Wolfe implementation, it has been long known that Fibonacci heaps have high constant-factor overheads that prevent their practical use [33, 34]. We still find the result useful in being the first to demonstrate a faster iteration speed, as well as a deterministic case to verify the correctness of our approach. For this reason we will use Algorithm 3 to show that our method converges at the same rate and with fewer FLOPs compared to the standard Frank-Wolfe implementation, as it does not suffer from the randomness required for differential privacy[3]. Our concern about this inefficiency is limited, as many faster algorithms exist for non-private LASSO regression that are orders of magnitude faster than using Frank-Wolfe [35, 36, 37], so other tools can suffice. However, no tools for high-dimensional and sparse DP LASSO regression exist except for the Frank-Wolfe method, which we make more efficient in the next section.

### 3.3  Algorithmically Efficient Differentially Private Frank-Wolfe

For our DP Frank Wolfe algorithm, we convert from the Laplacian mechanism originally used by Talwar et al. to the Exponential Mechanism [38]. Rather than adding noise to each gradient and selecting the maximum, in the exponential mechanism each coordinate $j$ is given a weight $\propto \exp\left(\frac{\epsilon u(j)}{2\Delta u}\right)$ where $u(j)$ is the score of the $j^{\text{th}}$ item and $\Delta u$ is the sensitivity [39].

This poses two challenges:

1. We need to select a weighted random sample from $D$ options in sub-$\mathcal{O}(D)$ time to pick the coordinate $j$ to maintain the $DP$ of the exponential mechanism.

2. We need to do this while avoiding the numeric instability of raising a gradient to an exponential, which will overflow for relatively small gradient magnitudes.

We tackle both of these issues by adapting the A-ExpJ algorithm of [40], and use their naming conventions to make comparison easier. This algorithm works on a stream of items with weights $w_i$, and in $\mathcal{O}(1)$ space can produce a valid weighted sample from the stream. It does so by computing a randomized threshold $T_w$ and processing samples until the cumulative weights $\sum_i w_i > T_w$, at which the final item in the sum becomes the new sample. A new value of $T_w$ is computed, and the process continues until the stream is empty. This process requires generating only $\mathcal{O}(\log D)$ random thresholds for a stream of $D$ items.

We exploit the fact that we have a known and fixed set of $D$ items to develop a new version of this algorithm that can be updated in constant time, is numerically sta-

---

**Algorithm 4** Big-Step Little-Step Exponential Sampler

1: **function** GETNEXT( )
2:    $j \leftarrow 0$
3:    $o \leftarrow \exp(\mathbf{v}^{(j)} - z_\Sigma)$
4:    $logT_w \leftarrow \frac{\log \mathcal{U}(0,1)}{\exp \mathbf{v}^{(i)} - z_\Sigma}$
5:    $c \leftarrow 0$
6:    **while** $c < N$ **do**
7:      $X_w \leftarrow \frac{\log \mathcal{U}(0,1)}{logT_w}$
8:      **while** $\exp\left(\mathbf{c}^{(c \bmod \lfloor\sqrt{N}\rfloor)} - z_\Sigma\right) - o < X_w$ **do**
9:        $X_w \leftarrow X_w - \left(\exp\left(\mathbf{c}^{(c \bmod \lfloor\sqrt{N}\rfloor)} - z_\Sigma\right) - o\right)$
10:        $o \leftarrow 0$
11:        $c \leftarrow \lfloor\sqrt{N}\rfloor - c \bmod \lfloor\sqrt{N}\rfloor$
12:      **end while**
13:      **while** $\exp\left(\mathbf{v}^{(c)} - z_\Sigma\right) < X_w$ **do** $\color{green}\mathcal{O}(\sqrt{N}\log N)$
14:        $X_w \leftarrow X_w - \exp\left(\mathbf{v}^{(c)} - z_\Sigma\right)$
15:        $o \leftarrow o + \exp\left(\mathbf{v}^{(c)} - z_\Sigma\right)$
16:        $c \leftarrow c + 1$
17:      **end while**
18:      **if** $c < N$ **then**
19:        $j \leftarrow c$
20:        $c \leftarrow c + 1$
21:        $t_w \leftarrow \exp\left(\exp\left(\mathbf{v}^{(j)} - z_\Sigma\right)\right) \cdot logT_w$
22:        $logT_w \leftarrow \frac{\log \mathcal{U}(t_w,1)}{\exp\left(\mathbf{v}^{(j)} - z_\Sigma\right)}$
23:        **if** $j \bmod \lfloor\sqrt{N}\rfloor \neq 0$ **then**
24:          $o \leftarrow 0$
25:        **else**
26:          $o \leftarrow o + \exp\left(\mathbf{v}^{(j)} - z_\Sigma\right)$
27:        **end if**
28:      **end if**
29:    **end while**
30: **end function**
31: **function** UPDATE($i, v$)
32:    $v_{cur} \leftarrow$ current priority of item $i$
33:    $k \leftarrow i \bmod \lfloor\sqrt{N}\rfloor$
34:    $\mathbf{c}^{(k)} \leftarrow \mathbf{c}^{(k)} + \log\left(1 - e^{v_{cur}-\mathbf{c}^{(k)}} + e^{v-\mathbf{c}^{(k)}}\right)$
35:    $z_\Sigma \leftarrow z_\Sigma + \log\left(1 - e^{v_{cur}-z_\Sigma} + e^{v-z_\Sigma}\right)$
36: **end function**

---

[3]We note that there can be mild disagreement on update order caused by numerical differences in brute-force recalculation of Algorithm 1 and the updates of Algorithm 2, but we observe no issues with this.

ble for a wide range of weights, and can draw a new sample in $\mathcal{O}(\sqrt{D}\log D)$ time. The key idea is to form large groups of variables and keeping track of their collective weight. If the group's weight is smaller than $T_w$, then the entire group can be skipped to perform a "Big-Step". If the group's weight is larger than $T_w$, then the members of the group must be inspected individually to form "Little-Steps". For this reason we term our sampler the "Big-Step Little-Step" sampler, and this procedure is shown in Algorithm 4.

The scale of gradients can change by four or more orders of magnitude due to the evolution of the gradient during training and the exponentiation of the Exponential Mechanism. For this reason, all logic is implemented at log scale, and a total log-sum-weight $z_\Sigma$ is tracked. Every exponentiation of a log-weight then subtracts this value, performing the log-sum-exp trick to keep the sample weights in a numerically stable range[4]

Similarly, each group has a group log-sum weight, and we denote the vector of the group weights as $\mathbf{c}$. There are $\sqrt{D}$ groups so that each group has $\sqrt{D}$ members. On lines 34 and 35 of Algorithm 4, a log-sum-exp update is used to update the group sum $\mathbf{c}^{(k)}$ and total sum $z_\Sigma$ (which are already log-scale since there was no exponentiation on line 30 of Algorithm 2). In both cases we always expect the group sum to be larger, and so we use $\mathbf{c}^{(k)}$ and $z_\Sigma$ as the maximal values to normalize by in each update. Lines 31 and 32 select the "Big-Step" group to update for the change in weight of the $i$'th item.

Lines 8-12 and 13-17 perform the same loop at two different scales. 8-12 perform big steps over groups, and must handle that the starting position could be in the middle of a group from a previous iteration, making it a partial group. For this reason, there is a "group offset" $o$ that subtracts the weight of items already visited in the group. Once a Big-Step is made, on line 11 the position is incremented by the group size modulo the current position, so that each step starts at the beginning of the next group regardless of starting position, handling the case of starting from a previous little step's location. Then lines 13-17 perform little steps within a group, and it is known that a new item must be found in the little group, otherwise, lines 8-12 would have repeated due to having a sum smaller than $T_w$.

The remainder of lines 2-5 and 18-30 work as the standard A-ExpJ algorithm, except each calculation, is done at log-scale or exponentiated if an item is needed at a non-log scale, for example on line 21[5].

Each of the $\log(D)$ random variates needed by Algorithm 4 corresponds to the selection of a new current sample. In the worst cases, each of these samples will belong to a different group, necessitating exploring $\mathcal{O}(\log D)$ groups each of size $\sqrt{D}$ by construction, giving a total sampling complexity of $\mathcal{O}(\sqrt{D}\log D)$. Just as with the Fibonacci Heap, the update procedure is $\mathcal{O}(1)$ per update, and so the final DP-FW complexity becomes $\mathcal{O}(NS_cT\sqrt{D}\log D + TS_rS_c)$. As we will demonstrate in our results, this provides significant speedups over the standard Frank-Wolfe for sparse datasets. This is because, by design, the Algorithm 4 procedure is very cache friendly, performing linear scans over $\sqrt{D}$ items at a time making pre-fetching very easy, and thus has only $\mathcal{O}(\log D)$ cache-misses when performing Little-Step transitions.

## 4 Results

Having derived a sparsity-aware framework for implementing Frank-Wolfe iterations in time proportional to the sparsity of the data, we will now demonstrate the effectiveness of our results. Since our goal is to support faster training with sparse datasets, we focus on high-dimensional

Table 2: Datasets used for evaluation. We focus on cases that are high-dimensional and sparse.

| Dataset | N | D |
|---|---|---|
| RCV1 | 20,242 | 47,236 |
| 20 Newsgroups.Binary "News20" | 19,996 | 1,355,191 |
| Malicious URLs, "URL" | 2,396,130 | 3,231,961 |
| Webb Spam Corpus, "Web" | 350,000 | 16,609,143 |
| KDD2010 (Algebra), "KDDA" | 8,407,752 | 20,216,830 |

problems listed in Table 2 where $D \geq N$. Note that to the best of our knowledge, the RCV1 dataset at

---

[4]Very small weights will still underflow, but by definition this happens when their probability to be selected is several orders of magnitude lower and thus are astronomically unlikely to be chosen anyway. Adding a small $10^{-15}$ value guarantees a chance to be selected and maintains DP via technically adding more noise than necessary.

[5]We note that the double exponentiation on this line is correct and numerically stable. The first exponentiation will produce a value in the range of $[0, 1]$ for the second exponentiation to use per the A-ExpJ algorithm

$D = 47$k is the highest-dimensional dataset any prior work has used to train a DP logistic regression model [16], and its $D$ is $428\times$ smaller than the largest $D$ we consider.

Our results will first focus on the non-private Frank-Wolfe due to its deterministic behavior and clear convergence criteria via the Frank-Wolfe gap $g_t$[6]. This will allow us to show clearly that we reduce the total number of floating point operations per second (FLOPs) required, though we note that in practice the runtime remains similar due to cache inefficiency.

After establishing that Algorithm 2 requires fewer FLOPs, we turn to testing the DP version leveraging our Big-Step Little-Step Sampler Algorithm 4, showing speedups ranging from $10\times$ to $2,200\times$ that of the standard DP Frank-Wolfe algorithm when training a model on sparse datasets.

Due to the computational requirements of running all tests, we fix the total number of iterations $T = 4,000$ and maximum $L_1$ norm for the Lasso constraint to be $\lambda = 50$ in all tests across all datasets. This value produces highly accurate models in all non-private cases, and the goal of our work is not to perform hyper-parameter tuning but to demonstrate that we have taken an already known algorithm with established convergence rates and made each iteration more efficient. All experiments were run on a machine with 12 CPU cores (though only one core was used), and 128GB of RAM. The total runtime for all experiments took approximately 1 week and exploring larger datasets was limited purely by insufficient RAM to load larger datasets in memory. Our code was written in Java due to the need for explicit looping, and implemented using the JSAT library [36]. When comparing Algorithm 1 and our improved Algorithm 2, the latter will be prefixed with "-Fast" in the legend to denote the difference.

## 4.1 Non-Private Results

We first begin by looking at the convergence gap $g_t$ over each iteration to confirm that we are converging to the same solution, which is shown in Figure 1. Of note, it is often impossible to distinguish between the standard and our fast Frank-Wolfe implementations because they take the exact same steps. Differences that occur are caused by nearly equal gradients between variables and are observable via inspection of $g_t$. In all cases, the solutions returned achieve identical accuracy on the test datasets.

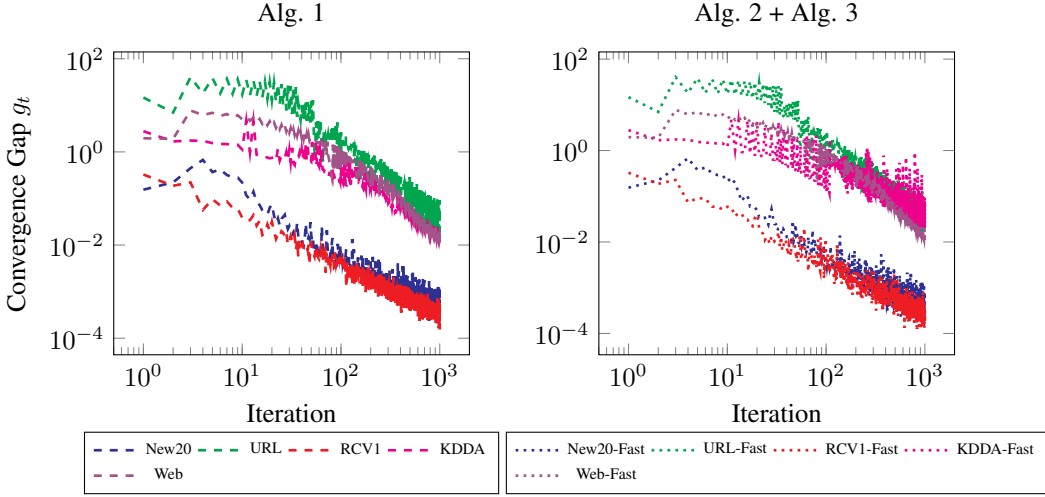

Figure 1: Convergence gap $g_t$ (y-axis, same scale for both plots) as the number of iterations increases (x-axis), showing that our Algorithm 2 (dotted lines) converges to the same solutions as Algorithm 1, with minor differences due to numerical floating point changes (i.e., both plots look nearly identical, the desired behavior). This shows our new approach maintains solution quality.

In Algorithm 2, the updating in differences can cause catastrophic cancellation due to the zig-zag behavior of Frank-Wolfe iterates (updating the same coordinate $j$ multiple times with differing

---

[6]In the DP case $g_t$ is especially noisy and hard to leverage meaningfully, often starting lower and increasing. This makes it a non-informative measure

signs on each update), resulting in similar magnitude sign changes that result in slightly different results numerically compared to re-computing the entire gradient from scratch. Choosing an adaptive stepsize $\eta_t$ may alleviate this issue in future work.

Next, we empirically validate the $\mathcal{O}(\|\mathbf{w}^*\|_0)$ number of times we must query the Fibonacci Heap for selecting the next iterate. The ratio of the number of times we must pop an item from the Heap against the value of $\|\mathbf{w}^*\|_0$ is plotted over training iterations in Appendix Figure 3. We can see in all cases the ratio is $\leq 3$, and so few calls to the heap are necessary.

Finally, we look at the convergence rate $g_t$ as a function of the number of FLOPs required to obtain it, as shown in Figure 2. It is clear that we avoid orders of magnitude more operations than a naïve implementation of Frank-Wolfe would normally require, providing the foundation for our faster DP results in the next section. Unfortunately, the Fibonacci Heap has poor caching behavior, resulting in no meaningful difference in runtime for the sparse case.

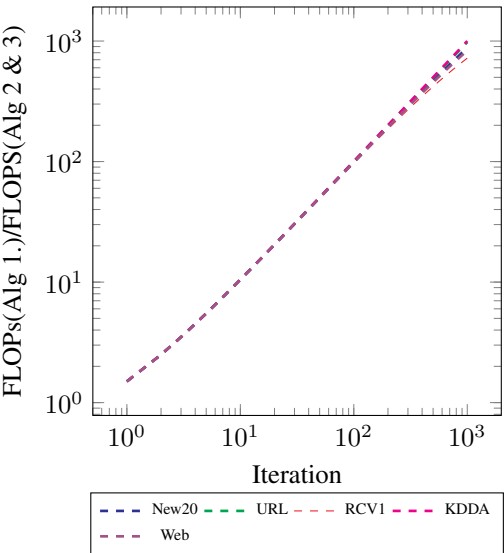

Figure 2: The y-axis (larger is better) shows how many times fewer FLOPs our Alg. 2 + Alg 3 needs compared to the original Frank-Wolf (Alg. 1). The x-axis is how many iterations into training have been performed. Note that in all cases the difference is difficult to differentiate reaching 1,000 iterations as we reduce the number of FLOPs by orders of magnitude per iteration.

## 4.2 Differentially Private Results

Having established that Algorithm 2 avoids many floating point operations by exploiting sparseness in the training data, we now turn to training DP versions of the normal and our faster Frank-Wolfe algorithms. The total speedup in the runtime of our Algorithm 2 over Algorithm 1 is shown in Table 3. As can be seen, our results range from a $10\times$ speedup for the URL dataset at the low end, up to a $2,200\times$ speedup for the KDDA and URL datasets at $\epsilon = 0.1$. In addition we ablated using Algorithm 2 with the brute force noisy-max as an ablation, which shows smaller speedups. This demonstrates that the combination of Algorithms 2 and 4 are necessary to get our results.

Table 3: How many times faster our Algorithm 2+Algorithm 4 over the standard FW implementation. In addition, just Algorithm 2 using the noisy-max sampling is included as an ablation, showing that both Alg. 4 & 2 combined are necessary to obtain maximum speed.

| Dataset | $\epsilon_1$ | | $\epsilon_{0.1}$ | |
|---|---|---|---|---|
| | Alg. 2+4 | Alg. 2 | Alg. 2+4 | Alg. 2 |
| News20 | 81.69 | 17.83 | 93.51 | 19.05 |
| URL | 9.99 | 1.02 | 2451.80 | 95.58 |
| RCV1 | 19.44 | 1.36 | 20.37 | 1.82 |
| Web | 581.25 | 21.24 | 537.65 | 20.79 |
| KDAA | 1239.64 | 206.50 | 2245.56 | 368.96 |

We note that the speedup of our method is a function of the sparsity of informative and non-informative features. This is more noticeable on the URL dataset which jumps from a $10\times$ speedup to a $2,400\times$ speedup when moving from $\epsilon = 1$ down to $\epsilon = 0.1$. This is because the URL dataset has 200 dense features that are highly informative, and the remaining features are all sparse. When a feature is dense, there is no advantage to using Alg. 2 & 4, and so no benefit is obtained. At the lower noise level of $\epsilon = 1$, the denser (and thus slower) informative features are selected more frequently, resulting in longer total runtime. As the noise increase with $\epsilon = 0.1$, the sparser non-informative features are selected more often, which reduces the average amount of work per update. This phenomena occurs in most datasets as denser features intrinsically have more opportunities to be discriminative, but is uniquely pronounced on the URL dataset.

This is ultimately a desirable property of our method, as large values of $\epsilon > 10$ are effectively not private, and so faster methods of non-private training should be used instead. Our Algorithm 2 with the Big-Step-Little-Step Sampler of Algorithm 4 will increase in its effective utility as the desired amount of privacy increases.

Table 4: Even at high privacy $\epsilon = 0.1$ we obtain non-trivial Accuracy and AUC on most datasets by using $T = 400,000$ iterations. Because the datasets are so high dimensional, we still obtain sparse solutions (rightmost column).

| Dataset | Accuracy (%) | AUC (%) | Sparsity (%) |
|---------|--------------|---------|--------------|
| RCV1    | 90.53        | 97.29   | 5.81         |
| News20  | 92.37        | 98.65   | 75.19        |
| URL     | 73.23        | 82.60   | 89.44        |
| Web     | 75.42        | 92.51   | 99.90        |
| KDDA    | 85.25        | 53.31   | 85.30        |

As our final test to highlight the utility and importance of our approach, we re-run each dataset using $\lambda = 5000$ with $T = 400,000$ iterations at a real-world useful $\epsilon = 0.1$. As shown in Table 4 this results in non-trivial accuracy and AUC for all datasets but KDDA, and is only possible by performing hundreds of thousands of training iterations. Iyengar et al. [16] show the best prior results at $\epsilon = 0.1$ for RCV1 64.2% accuracy, and in fact, we trail the non-private accuracy of 93.5% by only 3% points (note as well that their solution has 0% sparsity). This is made possible by simply performing far more iterations, which is computationally intractable with prior methods. We also note that we obtain significant sparsity on the higher dimensional datasets News20, URL, Web, and KDDA due to the fact that $T < D$ for each of them, and the Frank-Wolfe will by construction have a number of non-zero coefficients $\leq T$.

## 5    Conclusion

We have developed the first DP training procedure that, for a sparse input dataset, obtains a training complexity that is sub-linear in the total number of features $D$ at $\mathcal{O}(NS_c + T\sqrt{D}\log D + TS_rS_c)$. Testing on multiple high-dimensional datasets, we obtain up to $2,200\times$ speedup, with increasing efficiency as the value of $\epsilon$ decreases. Using this speed efficiency, we show non-trivial accuracy and privacy at $\epsilon = 0.1$ for $352\times$ more features than ever previously attempted and improve accuracy compared to prior work by an absolute 26.3%.

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

## A Additional Results

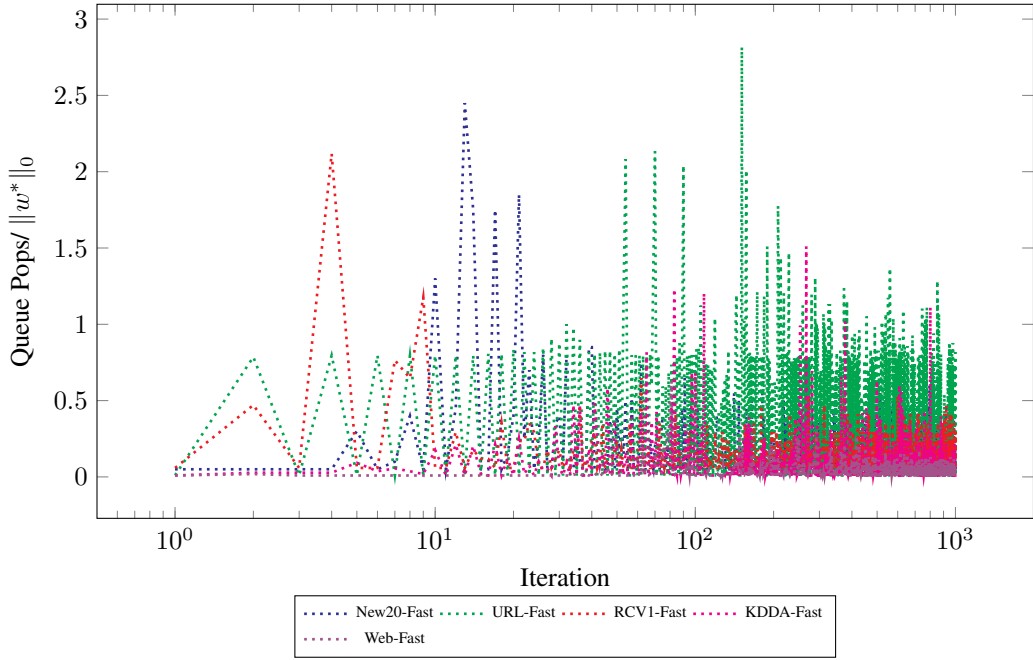

Figure 3: The ratio of items popped from a Fibonacci Heap over the non-zeros in the final solution (y-axis) over the number of iterations (x-axis) empirically demonstrates that Algorithm 3 does not need to consider all $D$ possible features to select the correct iterate.

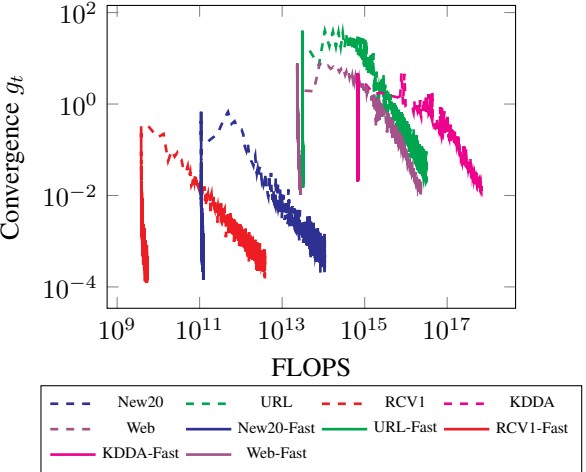

Figure 4: The number of floating point operations (x-axis) against the convergence gap $g_t$ (y-axis), showing that Algorithm 2 (solid lines) reduces the required numerical steps by multiple orders of magnitude over the original Frank-Wolfe Algorithm 1 (dashed lines).

## B Proofs of Correctness

Because our method is a direct translation of Frank-Wolfe to equivalent mathematical steps, all prior proofs apply to our version of the algorithm. We repeat/sketch the proofs here for further confidence in the correctness of the method.

## B.1 Utility/Error Bounds

We provide a proof sketch that our algorithm's utility is near-optimal below which follows that of [38].

Let $L$ be defined as in the paper: the Lipschitz constant of the loss function with respect to the $L_1$ norm. Let $S$ be the number of vertices of the constraint region, which we denote $\mathcal{C}$. Let $\lambda$ be the scaling factor of the $L_1$ ball to achieve the constraint region $\mathcal{C}$. Then for a loss with upper bound $\Gamma_{\mathcal{L}}$ on the curvature constant as defined in [41], running Algorithm 2 for $T = \frac{\Gamma_{\mathcal{L}}^{2/3}(n\epsilon)^{2/3}}{(L\lambda)^{2/3}}$ iterations, we have

$$\mathbb{E}[\mathcal{L}(\mathbf{w}^{priv}; D)] - \min_{\mathbf{w} \in \mathcal{C}} \mathcal{L}(\mathbf{w}; D) = \mathcal{O}\left(\frac{\Gamma_{\mathcal{L}}^{1/3}(L\lambda)^{2/3}\log(n|S|)\sqrt{\log(1/\delta)}}{(n\epsilon)^{2/3}}\right).$$

To prove this statement, we use Lemma 5 and Theorem 1 from [41]. Lemma 5 states that at each step of the Frank-Wolfe algorithm, an inexact gradient can be used so long as its score is within $\kappa$ to that of the true minimum vertex. [38] computed the value of $\kappa$ and showed that with probability $1 - \xi$, this occurs over all steps. Then Theorem 1 in [41] stated that if Lemma 5 holds, an exact bound on the overall loss holds. Thus it follows that with probability $1 - \xi$, this statement holds. Finally, [38] used standard learning theory arguments to convert the bound in probability to one in expectation. Plugging in the desired value of $T$ finishes the proof. Note that this proof is not affected by our sparse-aware framework. Indeed, the algorithm presented in [38] is simply algorithm [41] in our paper with dense calculations. (Lines 7-13 of our Algorithm 2 calculate scores for the exponential mechanism, like line 3 in Algorithm 2 of [38]. Line 15 in our algorithm corresponds to line 4 of Algorithm 2 in [38]. Lines 16-21 of our algorithm correspond to line 5 of Algorithm 2 in [38].) For that reason, at every iteration, as our updates are equivalent to that of [38].

Note that $\Gamma_{\mathcal{L}}$ can be upper bounded for logistic regression. See [17] for details.

Finally, using a fingerprinting codes argument, Theorem 3.1 of [38] showed that under weak conditions, an optimal DP-learning algorithm $\mathcal{A}$ has

$$\mathbb{E}[\mathcal{L}(\mathcal{A}(D); D) - \min_{\mathbf{w} \in \mathcal{C}} \mathcal{L}(\mathbf{w}; D)] = \widetilde{\Omega}\left(\frac{1}{n^{2/3}}\right).$$

For this reason, the utility bound provided above is nearly-optimal.

## B.2 Privacy Accounting

Here we prove that our algorithm is $(\epsilon, \delta)$-DP. First, note that the sensitivity of each update step is $\frac{L\lambda}{n}$, where $L$ is the Lipschitz constant of the loss function with respect to the $L_1$ norm and $\lambda$ is the scaling factor of the $L_1$ ball to achieve the constraint region $\mathcal{C}$. This is done using Lemma 2.6 from [42], in which we directly bound

$$|\langle s, \nabla\mathcal{L}(\mathbf{w}; D)\rangle - \langle s, \nabla\mathcal{L}(\mathbf{w}; D')\rangle|$$

where $s$ is any vertex of the $\mathcal{C}$. Now that we know the sensitivity, we can use the advanced composition theorem for pure differential privacy to find that $\epsilon = 2\epsilon'\sqrt{2T\log(1/\delta)}$. Rearranging, $\epsilon' = \frac{\epsilon}{\sqrt{8T\log(1/\delta)}}$. Thus, composing $T$ exponential mechanisms with privacy $\epsilon'$ produces a final result which is $(\epsilon, \delta)$-DP. In our algorithm, we use the Laplace distribution to implement the report noisy maximum version of the exponential mechanism at every iteration [43]. Thus our algorithm is $(\epsilon, \delta)$-DP.

