# OpenReview forum: "Scaling Up Differentially Private LASSO Regularized Logistic Regression via Faster Frank-Wolfe Iterations"
_NeurIPS.cc/2023/Conference — NeurIPS 2023 poster_

### Official Review · Reviewer_d4md · 2023-07-01

**Soundness:** 3 good
**Presentation:** 3 good
**Contribution:** 3 good
**Rating:** 6
**Confidence:** 3

**Summary:**

This paper studies how to equip differential priavate (DP) Frank Wolfe (FW) with the ability to cope with sparse data. The key insights are the proper priority queue data structures for handling the FW subproblem. To this end, a Fibonacci queue is proposed for non DP cases, and a big-step little-step sampler is designed for DP cases. The complexities of proposed approaches has improved dependence on N and D. Numerical results also suggests embracing sparsity fastens practical performance on large datasets.

**Strengths:**

(+) The proposed data structures are useful for handling sparse data for DP-FW. It provides a much better computation complexity with improved N and D dependence.

(+) Empirically, the smart use of these data structures leads to significantly faster performance that is up to 20x - 30x. Here I am comparing Alg. 2 + 4 with Alg. 2, since Alg. 2 is possibly a more reasonable benchmark if hopes to advance FW with sparsity.

(+) The numerical improvement tends to be more significant given a smaller $\epsilon$. This is helpful for settings with high requirement on privacy.

**Weaknesses:**

1.On the highlevel, this work focuses on developing efficient data structures for solving FW subproblem (in DP setting). Comparison with other data structures, such as those locality sensitive hashing ones https://arxiv.org/abs/2111.15139, is missing.

2.Can the proposed method benefit other FW approaches, such as https://arxiv.org/abs/2110.04243

3.The assertion in line 19 should be demonstrated more carefully.

**Questions:**

See weakness.

---

> ### Author Rebuttal · Authors · 2023-08-07
>
> Thank you for your key points that we will answer below and incorporate into the revision. We hope they satisfy your concerns, please let us know if we can further clarify anything.
>
> We respectfully note that our method is more than $20-30\times$ faster, as Alg. 2 is one of our contributions in this work. The purpose of Table 2 was to perform an ablation experiment to show that both Alg. 2 and Alg. 4 are important individually and together to obtain our speedups, which at a real-world privacy value of $\epsilon = 0.1$ range from $20\times$ up to $2451\times$ faster.
>
> >as those locality sensitive hashing ones https://arxiv.org/abs/2111.15139, is missing.
>
> Thank you for this highly related work we were unaware of. We will include it in the revision and related work for comparison and a more complete scope of literature.
>
> 2111.15139 tackles a very different setting from our own: they are focused on $N > D$, and non-private regression. We reached out to the authors during the rebuttal period who confirmed two further issues in performing a comparison.
>
> First, quoting the authors, "I believe it is possible to state that our [2111.15139] paper focused on the setting where d= log N", due to the LSH algorithm's domain of effectiveness being in situations where $D << N$.
>
> Second, the authors confirmed that no code was written for the paper and it is a theoretical work. Indeed, the manuscript does not state which of multiple possible data structures should be used for the maximum inner product search. While we made our own attempt in the time available, it does not yet work, and indeed, there are no baselines to compare against to know what is an expected speedup.
>
> Additionally, the paper is focused on the non-private case, meaning we would need to invent a new DP maxIPS data structure to use in our desired DP scenario. We suspect this alone would constitute a whole new paper of work and results.
>
> Other important differences include that the paper does not address sparsity in $D$, and more completely, their big-O complexity is $O(D + D N^\rho)$ where $\rho \in (0, 1)$ is a factor dependent on the maxIPS structure's efficiency on the current data. This thus does not tackle the $O(D)$ iteration cost we are concerned with in sparse data scenarios. In addition, if $T$ is the number of iterations needed to converge, their work proves they need $O(T/c^{2})$ iterations, an increase based on a maxIPS dependent constant $c \in (0, 1)$. The tradeoff between more iterations but faster per-iteration time would be another non-trivial factor.
>
> That said, the work is highly important in establishing a different approach to "queue maintenance" as we described in our work, by instead transforming the representation of the data, labels, and iterates, to accommodate the maxIPS data structures. They are also tackle the larger $N$ case instead of large & sparse $D$ of our own.
>
> We will synthesize these points into our revised related work and framing of our contribution.
>
> >benefit other FW approaches
>
> Our approach should be able to benefit many other FW approaches, but we can't state that it will work for _all_ other FW solvers. For the cited example of 2110.04243, we see that it should be a fairly direct application of our method to integrate with their own, with a few extra derivations of sparse updates for the added momentum term. We will cite and add this discussion to our related work, thank you.
>
> > line 19
>
> The reviewer's point is well taken. To avoid ambiguity we will revise Line 19 as a statement to ``all \textit{iterative} DP regression algorithms we are aware of''. We provide a table listing the training complexity of iterative private training procedures for high dimensional regression. Please note that the statement can be refined to $\mathcal{O}(TND)$ for non-sparse aware algorithms.
>
> | Method | Complexity |
> |:---:|:---:|
> | Frank-Wolfe Methods [1, 2, 3, 4] | $\mathcal{O}(TND)$ |
> | ADMM [5] | $\mathcal{O}(TNDM)$ |
> | Iterative Gradient Hard Thresholding Methods [4, 6, 7] | $\mathcal{O}(TND)$ |
> | Coordinate Descent [8] | $\mathcal{O}(TND)$ |
> | Mirror Descent [3] | $\mathcal{O}(TNDM)$ |
>
> Note that $M$ represents an iterative parameter which is greater than or equal to $1$. Methods with $M$ have a double-for loop and thus have two iterative parameters ($T$ and $M$).
>
> We thank the reviewer for this comment and will insert this table into the final version of our paper.
>
> [1] Talwar, Kunal, Abhradeep Guha Thakurta, and Li Zhang. "Nearly optimal private lasso." Advances in Neural Information Processing Systems 28 (2015).
>
> [2] Bassily, Raef, Cristóbal Guzmán, and Anupama Nandi. "Non-euclidean differentially private stochastic convex optimization." Conference on Learning Theory. PMLR, 2021.
>
> [3] Asi, Hilal, et al. "Private stochastic convex optimization: Optimal rates in l1 geometry." International Conference on Machine Learning. PMLR, 2021.
>
> [4] Hu, Lijie, et al. "High dimensional differentially private stochastic optimization with heavy-tailed data." Proceedings of the 41st ACM SIGMOD-SIGACT-SIGAI Symposium on Principles of Database Systems. 2022.
>
> [5] Wang, Puyu, and Hai Zhang. "Differential privacy for sparse classification learning." Neurocomputing 375 (2020): 91-101.
>
> [6] Wang, Lingxiao, and Quanquan Gu. "Differentially private iterative gradient hard thresholding for sparse learning." 28th International Joint Conference on Artificial Intelligence. 2019.
>
> [7] Wang, Lingxiao, and Quanquan Gu. "A knowledge transfer framework for differentially private sparse learning." Proceedings of the AAAI Conference on Artificial Intelligence. Vol. 34. No. 04. 2020.
>
> [8] Mangold, Paul, et al. "High-Dimensional Private Empirical Risk Minimization by Greedy Coordinate Descent." International Conference on Artificial Intelligence and Statistics. PMLR, 2023.

---

> > ### Comment · Reviewer_d4md · 2023-08-16
> >
> > Thank you for the response. It is a nice contribution and my score remains the same.

---

> > > ### Author Response · Authors · 2023-08-17
> > >
> > > We are glad we could satisfy the reviewer's questions and for your support of the paper. Please do not hesitate to let us know of any further questions we can clarify.

---

### Official Review · Reviewer_Vyxf · 2023-07-06

**Soundness:** 3 good
**Presentation:** 3 good
**Contribution:** 3 good
**Rating:** 6
**Confidence:** 3

**Summary:**

The paper presents an approach to train differentially private regression models on sparse input data. It leverages a modified version of the Frank-Wolfe algorithm to reduce the training time significantly by making the algorithm sensitive to sparse inputs.The algorithmic complexity is improved from the standard linear complexity to sub-linear. The authors have used their proposed method on multiple high-dimensional datasets improving the accuracy by 26.3% compared to previous methods.

**Strengths:**

The paper is eloquently written, with algorithms clearly delineated and accompanied by ample commentary. The proposed algorithm showcases its versatility across various scenarios, offering a remarkable improvement in performance.

**Weaknesses:**

The paper does not provide proven error bounds, and all theoretical results appear to revolve around the concept of speed optimization. It remains unclear whether this speed enhancement comes at the expense of accuracy. Additionally, the lack of released code poses a challenge to reproducibility.

Privacy accounting is implict and lacking, the authors didn't directly explain how does their algorithm satisfy the DP condition (advanced compostion?).

**Questions:**

Could you please clarify how the proposed algorithm ensures compliance with the Differential Privacy (DP) condition? Are there any alternative privacy accounting methods that may potentially enhance the overall performance? For example, better privacy accounting [1], using GDP/RDP [2] or tighter compostion theorems [3].

The authors suggest that their implementation requires fewer FLOPs compared to the standard Frank-Wolfe approach. Does this reduction in FLOPs translate directly to a commensurate speed increase, or is computational efficiency still constrained by factors such as RAM and cache access? Could you elaborate on the current computational bottlenecks?

The experimentation appears to have been conducted using a single core. Would there be any potential benefits, such as efficiency or performance improvements, if this method were to be adapted for multicore processing?

Can this method be extended beyond LASSO Regularized Logistic Regression? If not, what's the main challenge?

[1]: Altschuler, J., & Talwar, K. (2022). Privacy of noisy stochastic gradient descent: More iterations without more privacy loss. Advances in Neural Information Processing Systems, 35, 3788-3800.

[2] Liu, Y., Sun, K., Jiang, B., & Kong, L. (2022). Identification, amplification and measurement: A bridge to gaussian differential privacy. Advances in Neural Information Processing Systems, 35, 11410-11422.

[3] Kairouz, P., Oh, S., & Viswanath, P. (2015, June). The composition theorem for differential privacy. In International conference on machine learning (pp. 1376-1385). PMLR.

**Limitations:**

See weakness.

---

> ### Author Rebuttal · Authors · 2023-08-07
>
> > Could you please clarify how the proposed algorithm ensures compliance with the Differential Privacy (DP) condition?
>
> The original Alg. 1 has already been proven to be DP. Our work makes no changes that alter what is computed, and only avoids performing redundant calculations (i.e., multiplication by 0 gets 0), and thus all profs are still applicable. The reviewer's questions helped us realize this is not sufficient, see the below proof for additional confidence:
>
> Here we prove that our algorithm is $(\epsilon, \delta)$-DP.
>
> First, note that the sensitivity of each update step is $\frac{L\lambda}{n}$, where $L$ is the Lipschitz constant of the loss function with respect to the $L_1$ norm and $\lambda$ is the scaling factor of the $L_1$ ball to achieve the constraint region $\mathcal{C}$. This is done using Lemma 2.6 from [1], in which we directly bound
>
> $$\lvert \langle s, \nabla \mathcal{L}(\mathbf{w}; D) \rangle  - \langle s, \nabla \mathcal{L}(\mathbf{w}; D') \rangle \rvert$$
>
> where $s$ is any vertex of the $\mathcal{C}$. Now that we know the sensitivity, we can use the advanced composition theorem for pure differential privacy to find that $\epsilon = 2\epsilon'\sqrt{2T \log (1/\delta)}$. Rearranging, $\epsilon' = \frac{\epsilon}{\sqrt{8T \log (1/\delta)}}$. Thus, composing $T$ exponential mechanisms with privacy $\epsilon'$ produces a final result which is $(\epsilon, \delta)$-DP. In our algorithm, we use the Laplace distribution to implement the report noisy maximum version of the exponential mechanism at every iteration [2]. Thus our algorithm is $(\epsilon, \delta)$-DP.
>
> [1] Shalev-Shwartz, Shai. "Online learning and online convex optimization." Foundations and Trends® in Machine Learning 4.2 (2012): 107-194.
>
> [2] Bhaskar, Raghav, et al. "Discovering frequent patterns in sensitive data." Proceedings of the 16th ACM SIGKDD international conference on Knowledge discovery and data mining. 2010.
>
> >Are there any alternative privacy...
>
> Our algorithm works by composing a number of $(\epsilon', 0)$-DP steps to produce an $(\epsilon, \delta)$-DP algorithm with advanced composition, where $\epsilon'$ is chosen appropriately [4]. Composing $(\epsilon', 0)$-DP steps with the advanced composition theorem is tight [5].
>
> The works provided in this review address composition under different settings. [1] describes the setting in which privacy parameters are to be computed after training with Gaussian noise. This is common in DP deep learning systems. In our work, we set $(\epsilon, \delta)$ and $T$ prior to training, so accounting is not necessary. [2] describes Gaussian/Reyni differential privacy, which provides a tighter privacy composition when composing multiple $(\epsilon, \delta)$-DP algorithms. In our case, we compose multiple $(\epsilon, 0)$-DP algorithms, in which case the advanced composition theorem is tight. Finally, [3] describes a tight composition theorem for $(\epsilon, \delta)$-DP. Since wer are not composing $(\epsilon, \delta)$-DP steps, this is not necessary.
>
> We will ensure to cite these papers in our work and explain why using advanced composition to compose multiple $(\epsilon, 0)$-DP steps is tight.
>
> [4] Bhaskar, Raghav, et al. "Discovering frequent patterns in sensitive data." Proceedings of the 16th ACM SIGKDD international conference on Knowledge discovery and data mining. 2010.
>
> [5] Near, Joseph P., and Chiké Abuah. "Programming Differential Privacy." (2021).
>
>
> >Does this reduction in FLOPs translate directly
>
> The reviewer is correct that it is not a one-to-one translation in FLOPs reduction to speedup, we will make this more explicit in the revision.
>
> For the non-private case, _we obtain no speedups_, as the Fibonacci heap is highly cache inefficient. The numerous cache misses cause the program to hit the canonical "memory wall", meaning our throughput is completely IO bound on main-memory access. This is true of the standard FW in Alg. 1, and so has almost identical runtime. This was mentioned on line 372.
>
> We are not concerned about the lack of speedup in the non-private case, because much faster algorithms than FW already exist in this scenario (e.g., Liblinear is several hundred times faster in our testing).
>
> In the DP case, Alg. 1 is always compute bound, and our Alg. 2 + 4 combination vacillates between compute and memory bound on a per-iteration basis due to fairly atypical memory access patterns. Different dimensions have different sparsity patterns, resulting in cache misses, and different ratios of compute-to-memory accesses. This is the cause of higher speedups at lower $\epsilon$ we mentioned, due to avoiding the compute-bound by highly non-informative sparser features that are selected more often at high privacy.
>
> > ...if this method were to be adapted for multicore processing?
>
> Not directly, as many parts are memory bound and multiple threads polling memory would only increase this issue, while also adding synchronization overhead. Other prior works on parallelizing FW might be adaptable, but because we do less work, this is unclear as the ratio of work-per-thread will change dramatically.
>
> > Can this method be extended beyond [LR]?
>
> Yes, this method should work for any FW optimizable objective (LR, Linear Regression, Hinge-loss SVM, Gambler's loss, etc).

---

> > ### Comment · Reviewer_Vyxf · 2023-08-17
> >
> > Thank you for providing clarifications. I find that this paper presents a good contribution to regression methods under Differential Privacy. I have decided to raise my score accordingly.

---

> > > ### Author Response · Authors · 2023-08-18
> > >
> > > We are very glad we could satisfy your questions and appreciate the raised score!
> > >
> > > Please note that an error appears to have occurred with OpenReview and we can only see your original review/score apparently. If the AC could confirm that they see the correct version we would appreciate it.
> > >
> > > Thank you for your time and valuable feedback!

---

### Official Review · Reviewer_Fgow · 2023-07-07

**Soundness:** 2 fair
**Presentation:** 2 fair
**Contribution:** 2 fair
**Rating:** 6
**Confidence:** 3

**Summary:**

This paper proposes new private variants of the Frank-Wolfe algorithm that takes advantage of the sparsity in the data. The proposed methods are shown to significantly speed up Frank-Wolfe algorithm in a multitude of tasks while still achieving similar accuracy.

**Strengths:**

- The authors make an effort to explain a lot of what is going on in their algorithm.

- Since the result of the paper is about reducing computational complexity, I appreciate that the author denotes the complexity of every operation in their algorithms.

- The proposed method seems to do well in practice. The speed-up is quite significant.

**Weaknesses:**

- Even though I appreciate the effort the author put in to explain their algorithms, I still feel like the explanation is very confusing. The paper would really benefit if the author can add a bit more discussion in the appendix to make things a bit clearer.

- Some of the technical terms are used in the paper before being defined properly. For example, the term FLOPs is used in line 187 but only is defined in line 319.  Or the Fibonacci Heap is used without even having a few sentences defining what it is. Again, I think the author should consider having more discussions in the appendix.

- Section 3.3 presents a private algorithm without having a Theorem showing that the algorithm is $\epsilon$-DP.

- Figure 1 and 2 are very confusing. For example, in the caption of Figure 1, it says that Algorithm 2 is the dotted line but every line is a dotted line. The legend of the chart also does not have either Algorithm 2 or Algorithm 1.

- The authors use the phrase "handle something sparsely" often but it is a bit unclear what it means. Maybe explain it in the beginning or say something like "allow us to handle the update in 1 dimension"?

**Questions:**

- In Algorithm 1, what do we need $g_t$ for? It isn't used anywhere else in the algorithm. Based on the experiment, seems like the authors want to output $g_t$ as the convergence gap, then the algorithm should output both $w_T$ and $g_t$?

- I'm a bit confused about the statement in line 139. Isn't $\bar v_t^{(i)} = \sum_{j=1}^dX^{i,j}w_t^{j}$. Thus, when 1 single coordinate of $w_t$ changes, every $\bar v_t^{(i)}$ will also change?

- Can the author explain how using $\gamma$ to update allow us to not use $\bar y$ anymore?

- What are $\textbf{c}$ and $z_\Sigma$ in Algorithm 4? Those 2 are used without being defined.

- What is the sparsity of the data used in the experiments?

---

> ### Author Rebuttal · Authors · 2023-08-07
>
> We believe, if accepted for the conference, the historical extra camera-ready page will greatly improve the readability of the manuscript and allow us to have more detailed exposition. Please see below for the answers to your questions, which we will incorporate into the revision.
>
> > what do we need $g_t$ for?
>
> $g_t$ is a measure of the convergence gap. Once $g_t = 0$, the algorithm has completely converged at the function minima. Use of $g_t$ is desirable in non-DP settings to confirm that a sufficiently good quality solution has been reached, and to confirm that the implementation is indeed converging. As Figure 1 shows, we converge to the solution at the same rate as the original dense Frank-Wolf formation. In the DP case, $g_t$ becomes noisy and is not as useful.
>
> >line 139
>
> We must be precise about the sparsity patterns. Every $\bar v_t^{(i)}$ **where** $X^{i,j} \neq 0$, will change. Because most values are equal to 0, the majority of $\bar v_t^{(i)}$ will not change. Otherwise, the reviewer is correct in the definition. To restate, only one value of $j$ in $\boldsymbol{w}^{(j)}$ will change each iteration (the others are handled by our scaling factor). In most high-dimensionsal regression problems (like all the ones we have used), most coordinates $j$ are unused by most columns $i$, and so the update of $\boldsymbol{\bar v}$ can be done in a sparse fashion.
>
> >how using  $\gamma$  to update allow us to not use  $\bar y$ anymore
>
> Note that in Algorithm 1, the values of $\boldsymbol{\bar y}$ are fixed during each iteration, and do not change. $\boldsymbol{\bar y}$ only impacts the algorithm by offsetting the values stored in $\boldsymbol{\bar \alpha}$. Because Alg. 2 updates $\boldsymbol{\bar \alpha}$ sparsely for the amount of change that occurs, and the value of $\boldsymbol{\bar y}$ will never change, then we can ignore the $\boldsymbol{\bar y}$ factor after the initialization of line 12 in Alg. 2.
>
> >What are $\boldsymbol c$  and $z_\Sigma$ in Algorithm 4?
>
> $\boldsymbol c$ was defied on lines 280-281. There are $\lfloor \sqrt{D} \rfloor$ groups of variables in Alg 4, and so $\boldsymbol c$ is a vector of size  $\lfloor \sqrt{D} \rfloor$. The $j$'th value $\boldsymbol{c}^{(j)}$ contains the cumulative weight of all variables contained in the $j$'th group. It is used to skip the $j$'th group if it's cumulative weight is smaller than the current step size, allowing us to perform a "Big Step".
>
> $z_\Sigma$ was defined on line 282. It is the cumulative weight of all variables being sampled from (i.e., $z_\Sigma = \mathop{LogSumExp}( \boldsymbol{c}) = \log \left( \sum_{j=1}^{\lfloor \sqrt{D} \rfloor}  \exp\left(\boldsymbol{c}^{(j)}\right) \right)$. It is necessary as the normalizing constant so that we ensure our sample is a proper weighted uniform random sample.
>
> Please note Alg. 4 uses $N$ instead of $D$ to match the notation of the original A-ExpJ paper, since the page limit of NeurIPS prevents us from a more thorough explanation in text.
>
> >What is the sparsity of the data used in the experiments?
>
> These datasets are highly sparse, see the below table for % of non-zero values.
>
> | Dataset | % of non zero values |
> |---|---|
> | RCV1 | 1.5% |
> | News20 | 0.03% |
> | URL | 0.004% |
> | Web | 0.022% |
> | KDDA | 0.00018% |
>
> >add a bit more discussion in the appendix
>
> We will add a brief explanation and intro in the appendix for each major section to ensure the accessibility of the work.
>
> > used in the paper before being defined
>
> We scoped our assumption of background too narrowly, and we will go through the paper again to ensure this does not remain in the revised manuscript.
>
> FLOP: is *F*loating *P*oint *Op*erations, normally taken to be the number of multiplications, divisions, and other more expensive functional primitives (e.g., computing the $\exp$ is a single instruction in modern hardware). Counting the FLOPs provides a standardized and hardware-independent way of quantifying the amount of expensive computations being performed, as floating-point operations are generally many times more expensive than other integer operations.
>
> Fibonacci Heap: A classic heap data structure for inserting, finding-the-minimum, and removing-the-minimum value from the data structure. The Fibonacci heap is relatively unique in supporting a decrease-key function that allows altering the value of an item already in the heap. All operations on a Fibonacci heap can be performed in amortized $O(1)$ time, with the exception of removal that takes $O(\log n)$ time. Using the decrease-key operation allows us to devise the queue maintenance strategy of Algorithm 3.
>
> >a private algorithm without having a Theorem
>
> Please see our reply to reviewer LVpw, which provides a proof. Our method is mathematically equivalent to the original Alg 1., and so all proofs of Alg 1 still apply to Alg 2 + 4, but reviewer feedback has made it clear that making this more explicit would aid in understanding the equivalence and validity.
>
> >Figure 1 and 2 are very confusing
>
> Please see the 1-page attachment the adds new versions of these figures, we hope it clarifies your concern.
>
> > handle something sparsely
>
> We will revise the manuscript to make explicit that what we mean by this is: if the original data $X^{i, j}$ has a value equal to zero, then as few operations as possible should occur involving the $i$ and $j$ values as the zero values will have no impact on the solution. We do not prove a minimum number of operations, but show for the first time it is sub $O(D)$ via the existence of our method.

---

> > ### Comment · Reviewer_Fgow · 2023-08-19
> >
> > Thanks for the response! My questions are pretty well-addressed. I will raise the score to 6.

---

> > > ### Author Response · Authors · 2023-08-20
> > >
> > > We are glad we are able to satisfy your questions, and very appreciative of the score raise! Please let us know if there is anything else that comes to mind.

---

### Official Review · Reviewer_LVpw · 2023-07-07

**Soundness:** 3 good
**Presentation:** 3 good
**Contribution:** 2 fair
**Rating:** 5
**Confidence:** 2

**Summary:**

This paper studies DP regression problem when the data is $\ell_1$-sparse and aims to improve the computational efficiency. Specifically, they consider LASSO regularized logistic regression model and sparse aware Frank-Wolfe algorithm. The proposed method computes mathematical equivalent results as in the original algorithms but computationally much more efficient for large sparse datasets. The experiments confirm that the proposed methods are significant faster than existing methods.

**Strengths:**

1. This paper provides a practical tool for DP regression on large sparse datasets. Prior works on sparse regressions are either purely theoretical or non-scalable. As far as I know, this is the first paper that focus on the computational efficiency.
2. This paper is well-written and clearly states its contribution.

**Weaknesses:**

This paper does not discuss any utility guarantees in terms of its theoretical dependence on $d, \lambda, L, \varepsilon, n$. I am uncertain about the performance of sparse-aware DP Frank-Wolfe on this problem. Previous work [1] is known to be near-optimal in terms of utility. If the proposed method is sub-optimal, accelerating a sub-optimal algorithm may be of less interest.

[1] Cai, T.T., Wang, Y., and Zhang, L. (2021). The cost of privacy: Optimal rates of convergence for parameter estimation with differential privacy. The Annals of Statistics, 49(5), 2825-2850.

**Questions:**

Please see weakness

**Limitations:**

This paper has not addressed the limitations.

---

> ### Author Rebuttal · Authors · 2023-08-07
>
> We should have made the DP bounds more explicit, thank you for bringing this to our attention. Please note that our algorithm's utility is near-optimal. We provide a proof sketch below which follows that of [1].
>
> Let $L$ be defined as in the paper: the Lipschitz constant of the loss function with respect to the $L_1$ norm. Let $S$ be the number of vertices of the constraint region, which we denote $\mathcal{C}$. Let $\lambda$ be the scaling factor of the $L_1$ ball to achieve the constraint region $\mathcal{C}$. Then for a loss with upper bound $\Gamma_{\mathcal{L}}$ on the curvature constant as defined in [2], running Algorithm 2 for $T = \frac{\Gamma_{\mathcal{L}}^{2/3}(n\epsilon)^{2/3}}{(L\lambda)^{2/3}}$ iterations, we have
>
> $$\mathbb{E}[\mathcal{L}(\mathbf{w}^{priv}; D)] - \min_{\mathbf{w} \in \mathcal{C}} \mathcal{L}(\mathbf{w}; D) = \mathcal{O} \left( \frac{\Gamma_{\mathcal{L}}^{1/3}(L\lambda)^{2/3}\log(n \lvert S \rvert) \sqrt{\log (1/\delta)}}{(n\epsilon)^{2/3}} \right).$$
>
> To prove this statement, we use Lemma 5 and Theorem 1 from [2]. Lemma 5 states that at each step of the Frank-Wolfe algorithm, an inexact gradient can be used so long as its score is within $\kappa$ to that of the true minimum vertex. [1] computed the value of $\kappa$ and showed that with probability $1 - \xi$, this occurs over all steps. Then Theorem 1 in [2] stated that if Lemma 5 holds, an exact bound on the overall loss holds. Thus it follows that with probability $1 - \xi$, this statement holds. Finally, [1] used standard learning theory arguments to convert the bound in probability to one in expectation. Plugging in the desired value of $T$ finishes the proof. Note that this proof is not affected by our sparse-aware framework. Indeed, the algorithm presented in [1] is simply algorithm [2] in our paper with dense calculations. (Lines 7-13 of our Algorithm 2 calculate scores for the exponential mechanism, like line 3 in Algorithm 2 of [1]. Line 15 in our algorithm corresponds to line 4 of Algorithm 2 in [1]. Lines 16-21 of our algorithm correspond to line 5 of Algorithm 2 in [1].) For that reason, at every iteration, as our updates are equivalent to that of [1].
>
> Note that $\Gamma_{\mathcal{L}}$ can be upper bounded for logistic regression. See [3] for details.
>
> Finally, using a fingerprinting codes argument, Theorem 3.1 of [1] showed that under weak conditions, an optimal DP-learning algorithm $\mathcal{A}$ has
>
> $$\mathbb{E}[\mathcal{L}(\mathcal{A}(D); D) - \min_{\mathbf{w} \in \mathcal{C}} \mathcal{L}(\mathbf{w}; D)] = \widetilde{\Omega} \left( \frac{1}{n^{2/3}} \right).$$
>
> For this reason, the utility bound provided above is nearly-optimal.
>
> [1] Talwar, Kunal, Abhradeep Guha Thakurta, and Li Zhang. "Nearly optimal private lasso." Advances in Neural Information Processing Systems 28 (2015).
>
> [2] Jaggi, Martin. "Revisiting Frank-Wolfe: Projection-free sparse convex optimization." International conference on machine learning. PMLR, 2013.
>
> [3] Khanna, Amol, Fred Lu, and Edward Raff. "Sparse Private LASSO Logistic Regression." arXiv preprint arXiv:2304.12429 (2023).

---

> > ### Comment · Reviewer_LVpw · 2023-08-15
> >
> > Thank you for your detailed response. It would be great if you could add such a discussion to the revision.

---

> > > ### Author Response · Authors · 2023-08-15
> > >
> > > We will absolutely be including this discussion in the revision. Your feedback and the other reviewers have helped us to realize the DP proofs needed further elaboration than what our original manuscript presented. We believe the final version will be a much stronger article because of it.
> > >
> > > We hope this has satisfied all of your concerns, please let us know if there are any outstanding questions.

---

### Author Rebuttal · Authors · 2023-08-07

Per the rebuttal instructions, we could include one page of a PDF file with any new plots or figures. Toward Reviewer Fgow's concern, we have a new version of Figure 1 and Figure 2 in the attached PDF, displayed as Figure 4 & 5 in the PDF file.

Figure 4 replaces Figure 1, and we plot both the normal FW (Alg 1.) and our improvement in the non-DP case (Alg 2 + 3) side by side, showing that they are identical, which is the desired outcome. We have changed none of the underlying mathematics of the FW algorithm, and so we converge to the same solutions (barring floating-point differences).

Figure 5 replaces Figure 2. We plot how many times fewer cumulative FLOPs are performed by iteration $T$ on the x-axis. At $T=1$ we perform basically the same number of flops due to initial setup, but each subsequent iteration performs sparse updates that need far fewer FLOPs, resulting in multiple orders of magnitude reduction in FLOPs as the number of iterations increases.

---

### Decision · Program_Chairs · 2023-09-21

**Decision:**

Accept (poster)

**Comment:**

This paper studies LASSO regularized logistic regression with differential privacy and assumes that the data is
L1-sparse. The paper designs a more computationally efficient algorithm based on Frank-Wolfe, The experiments show that the proposed methods are significantly faster than existing approaches and, in particular, are practical even at large scale.